# Evaluating the Difference in Neuropsychological Profiles of Individuals with FASD Based on the Number of Sentinel Facial Features: A Service Evaluation of the FASD UK National Clinic Database

**DOI:** 10.3390/children10020266

**Published:** 2023-01-31

**Authors:** Bethany M. Webster, Alexandra C. S. Carlisle, Alexandra C. Livesey, Lucy R. Deeprose, Penny A. Cook, Raja A. S. Mukherjee

**Affiliations:** 1Fetal Alcohol Spectrum Disorders (FASD) Specialist Clinic, Gatton Place, St Matthews Rd, Redhill, Surrey RH1 1TA, UK; 2School of Health and Society, University of Salford, Salford M6 6PU, UK

**Keywords:** FASD, facial features, neuropsychological profile

## Abstract

(1) It might be implied that those with Fetal Alcohol Spectrum Disorder (FASD) with fewer sentinel facial features have a “milder” neuropsychological presentation, or present with fewer impairments than those with more sentinel facial features. The aim of this service evaluation was to compare the neuropsychological profile of people with FASD with varying numbers of sentinel facial features. (2) A clinical sample of 150 individuals with FASD, aged between 6 and 37 years, completed various standardised assessments as part of their diagnostic profiling. These included the documented level of risk of prenatal alcohol exposure (4-Digit Diagnostic Code), sensory needs (Short Sensory Profile), cognition (Wechsler Intelligence Scale for Children—4th Edition; WISC-IV), and communication and socialisation adaptive behaviours (Vineland Adaptive Behavior Scale—2nd Edition; VABS-II). As FASD has high comorbidity rates of Autism Spectrum Disorder (ASD) and Attention Deficit Hyperactivity Disorder (ADHD), these were also reviewed. The profiles of the ‘FASD with 2 or 3 sentinel facial features’ group (*n* = 41; 28 male, 13 female) were compared with the ‘FASD with 0 or 1 sentinel facial features’ group (*n* = 109; 50 male, 59 female) using Chi² tests, independent sample *t*-tests, and Mann-Whitney U analyses (where appropriate). (3) There were no significant differences between the two comparison groups across any measure included in this service evaluation. (4) Whilst sentinel facial features remain an important aspect in recognising FASD, our service evaluation indicates that there is no significant relationship between the number of sentinel facial features and the neuropsychological profile of people with FASD in terms of severity of presentation.

## 1. Introduction

Fetal alcohol spectrum disorder (FASD) is a diagnostic term used in the UK since 2019. This relates to the impact of prenatal alcohol exposure on the brain and body and is a lifelong disability [1]. Studies from the USA and Europe indicate between 1–10% of children in the general population have FASD, with a UK population-based birth-cohort prevalence study indicating 6% of children screened positive for FASD [2]. A recent UK active case ascertainment study indicated rates between 1.8 and 3.8% [3].

The National FASD Specialist Clinic, based in the UK, uses the Scottish Intercollegiate Guidelines Network SIGN 156 diagnostic guidelines [4] to assess FASD, as these have been adopted by the National Institute of Clinical Excellence. As part of the National FASD Specialist Clinic assessment, four key areas are examined: prenatal alcohol exposure, sentinel facial features, central nervous system (CNS) impairments, and ruling out other possible causes for presenting difficulties.

According to the SIGN 156 guidelines, not all three sentinel facial features of FASD (palpebral fissure length < 3rd percentile, smooth philtrum, thin upper lip) are required to be present for a diagnosis of FASD to be received [4]. If all three are present, a diagnosis of ‘FASD with sentinel facial features’ is given (previously named fetal alcohol syndrome, according to 4-Digit diagnostic criteria [5]). Without all three, a diagnosis of ‘FASD without sentinel facial features’ is given. According to the SIGN 156 [4] and the Canadian Diagnostic guidelines [6], the number of sentinel facial features is the only thing differentiating between a diagnosis of ‘FASD with sentinel facial features’ and ‘FASD without sentinel facial features’. Other factors (such as level of prenatal alcohol exposure and CNS impairments) are not implicated in the ‘with or without sentinel facial features’ part of the diagnostic terminology.

There are ten CNS areas that can be included in an FASD assessment as highlighted in SIGN 156, which are as follows: brain structure; motor skills; cognition; language; academic achievement; memory; attention; executive functioning; affect regulation; and social communication, social skills, and adaptive functioning [4]. While there is extensive research on how individuals with FASD may be impaired in these areas [7], the research that has been conducted has shown inconsistencies. Whilst it is clear those with sentinel facial features have severe cognitive presentations [8], the nature of the presentation for those without facial features remain variable, leading to the anecdotal assumption that not having facial characteristics implies a milder presentation.

There have been various suggestions in previous literature why those with all three sentinel facial features present more ‘severely’, which range from a true effect through to recruitment bias based on the sample population [9,10]. However, this assumption is not consistent with the anecdotal impression that was gleaned from clinicians in the National FASD Specialist Clinic since its inception and from over a decade of seeing complex cases.

The present service evaluation carried out by the National FASD Specialist Clinic explores whether there are differences in the profile of CNS deficits based on the varying numbers of sentinel facial features. The specific CNS deficits compared are cognition, adaptive functioning, and social communication. Additionally, we compare the documented level of risk of prenatal alcohol exposure, sensory impairments, and prevalence of comorbid diagnoses of autism spectrum disorder (ASD) and attention deficit hyperactivity disorder (ADHD).

## 2. Methods

As primarily a retrospective service evaluation, the exploration has been registered and approved with Surrey and Borders Partnership NHS Foundation Trust clinical governance procedures (AU/002/10/2021). Service user/parental consent was obtained for each individual attending the National FASD Specialist Clinic for their anonymised data to be used in such evaluations to aid learning and to develop the clinic further. From all the cases seen in the National FASD Specialist Clinic since its origin in 2009, only a few cases declined to allow their data to be used for such purposes and were not included. Specific numbers for those declining are not held in the database. The database is ongoing and continues to develop.

### 2.1. The Clinic

The National FASD Specialist Clinic is a highly specialised multidisciplinary assessment clinic for people who have suspected FASD. Referrals are received from all parts of the UK and assessments are funded by the NHS or on an individual case-by-case basis. The clinic completes a comprehensive multidisciplinary assessment to diagnose FASD and comorbid presentations, as well as providing management recommendations to individuals and/or parents/carers [11]. The assessment involves a 2-day process with an approximate 3-month gap between assessment days. A wide range of measures and evaluations are conducted to triangulate information and reach conclusions about diagnostic thresholds. Comorbid diagnostic features alongside wider understanding of functioning is used to guide better management. Details of the National FASD Specialist Clinic and assessment process have been explained numerous times [11,12]. The clinic is one of the few specialist places in the UK that can reliably and accurately diagnose FASD, using set diagnostic standards, without sentinel facial features being present.

Assessments used at the National FASD Specialist Clinic have progressed from its origin in 2009 and have been updated as developments occurred. These changes over the years mean there are some discrepancies with how many individuals completed the assessments included in this service evaluation (Table 1). Certain assessments including: 4-Digit Diagnostic Code [5], WISC-IV [13], and ADHD Screening Questionnaire (developed by the clinic, based on the DSM-IV) were gathered during the face-to-face assessments. The WISC-IV was administered by an experienced Clinical Psychologist; the other listed face-to-face assessments, including identifying facial features, were conducted by an FASD Specialist Clinician. The SCQ [14], SSP [15], and VABS-II [16] are caregiver reports completed outside of the face-to-face assessment period to provide additional information. Details regarding the normative scores for these assessments can be obtained from their respective assessment manuals. As this was a clinical sample, not all measures were completed by all participants. This was for various reasons, for example, someone declining to participate on the assessment day.

### 2.2. Diagnostic Terminology

Diagnostic terminology of FASD in the National FASD Specialist Clinic has changed in line with updated guidance [4,6]. Fetal alcohol syndrome (FAS) is now referred to as FASD with sentinel facial features. Partial fetal alcohol syndrome (pFAS), or alcohol related neurodevelopmental disorders (ARND) are now regarded as FASD without sentinel facial features. Prior to 2016, the National FASD Specialist Clinic used the 2005 Canadian guidance [17]. A previous audit from the clinic showed good correlation between the diagnostic terminology and criteria between different approaches [18].

Facial features are currently assessed in the National FASD Specialist Clinic using both 2D photographic analysis using the 4-Digit approach [5], as well as 3D photographic evaluation. As the 3D evaluation is relatively new, and still under development, the analysis here is based solely on the assessment of the 2D images. The 4-Digit facial photographic approach codes the images in a four-item rank of 1–4 as follows: (1) no features, (2) mild features (at least one of the three sentinel facial features being present), (3) moderate features (at least two of the three sentinel features being present), (4) severe (all three of the expected three sentinel features being present). This categorisation does have some variability, as shown in the 4-Digit manual, but the examples above are the most common [5]. When combined with findings on the Brain, Growth, and Alcohol domains, the following diagnostic terms are most commonly seen. A score of 4 is often linked in the 4-Digit score to fetal alcohol syndrome, 3 with partial FAS, and a score of 2 or below with static encephalopathy (equivalent to current diagnosis of FASD without sentinel features) [5].

A score of 4 is equivalent to three sentinel facial features and is required to meet diagnostic criteria for FAS or FASD with sentinel facial features [5]. Due to the rarity of this presentation in our sample, relatively few cases with all three sentinel facial features have been seen in the clinic over 10 years. This would be consistent with findings from McQuire et al. (2019) that those presenting with two or all three sentinel facial features in their larger UK cohort were also rare (1.6%) [2]. The Institute of Medicine (IOM) Guidance published in 2016 [19] allow for the presence of at least two or three facial features for the diagnosis of FAS or pFAS. As such, further analysis was conducted to compare those with all three sentinel facial features to those with two sentinel facial features. This allowed us to decide if it would be possible to combine them into one larger ‘dysmorphic’ group. This would offer greater numbers of people in the ‘dysmorphic’ group (two or more sentinel facial features) and allow more direct comparison between those with greater and fewer dysmorphic features.

Therefore, four groups were initially created for analysis. Based on the 4-Digit rank [5], the groups were analysed as follows: group A; those with all three sentinel facial features (4), group B; those with two sentinel facial features (3), group C; a combined group of two or more (3 + 4), and group D; those with one or fewer features (1 + 2). Subject to there being no difference between group A and B for the main comparison and translating this to SIGN guidance [4], the main comparison was between those with two or more sentinel facial features (2–3 sff), group C, and those with one or fewer sentinel facial features (0–1 sff), group D.

### 2.3. Data Analysis

The assessment scores included in this service evaluation (Table 1), were extracted from a pre-existing SPSS database, with anonymised patient data. Overall, the present sample consisted of 150 service users (78 male and 72 female). The first 150 service users who had confirmed diagnosis of FASD were included for analysis in the present service evaluation.

All analyses were carried out using IBM SPSS Statistics 28. There were no significant differences or associations between those who had two or three facial features (i.e., scoring 3 or 4 on the 4-Digit approach), in relation to age, sex, or IQ. Therefore, we were confident to include both groups in the 2–3 sff comparison group.

Descriptive statistics for age and sex were carried out for both comparison groups. Chi² tests were performed on the following: sex, ADHD diagnosis, ASD diagnosis, and SSP. Based on the distribution of the data, either an independent-samples *t*-test or Mann–Whitney U test were used on the following: age, WISC-IV, SCQ, and VABS-II. The Holm–Bonferroni correction for t-tests and Mann–Whitney U tests were calculated in Microsoft Excel [20] to maintain the family-wise type I error rate for each of the *t*-tests.

## 3. Results

A total of 41 individuals (28 male, 13 female) were included in the 2–3 sff group, aged between 6 and 37 years (Mdn=9, IQR=7). The 0–1 sff group consisted of 109 individuals (50 male, 59 female), aged between 6 and 26 years (Mdn=12, IQR=6). There was a significant difference between the comparison groups in terms of sex (Table 2). The age of those 2–3 sff and 0–1 sff was compared using Mann–Whitney U test; the median age of the 0–1 sff was significantly higher than the median age of the 2–3 sff; however, upon Holm–Bonferroni correction, this was no longer significant.

For categorical data (ASD diagnosis, ADHD diagnosis, 4-Digit score, SSP), frequency and percentages are reported in Table 2. For continuous data (SCQ total score, VABS-II, WISC-IV), means, medians, standard deviations and interquartile ranges are reported in Table 3.

Table 2 shows the breakdown of comorbid diagnoses of ASD and ADHD. Most of the 2–3 sff (77.5%) and 0–1 sff (72.6%) groups received a comorbid diagnosis of ASD. Similarly, most of the 2–3 sff and 0–1 sff groups received a comorbid diagnosis of ADHD (82.9% and 80.6%, respectively). Table 2 also shows that there was no significant association between having FASD 2–3 sff or 0–1 sff, and neurodevelopmental outcomes, specifically comorbid diagnoses of ASD, and the type of ADHD diagnoses. With the SCQ total score, Table 3 shows that, on average, both the 2–3 sff and 0–1 sff groups scored above the suggested cut-off of 15 (mean scores of 16.44 and 16.07, respectively), which suggests a positive screen for ASD, as per the manual [14]. Table 3 shows that there was no significant difference between the 2–3 sff and the 0–1 sff groups in terms of the SCQ total score.

Table 2 shows that 51.2% of the 2–3 sff and 57.8% of the 0–1 sff had a high documented level of risk of prenatal alcohol exposure, with the rest being strongly suspected based on lifestyle information gathered. However, the details during pregnancy may have not had complete information. Additionally, Table 2 shows that there was no significant association between the 2–3 sff and 0–1 sff comparison groups and the documented level of risk of prenatal alcohol exposure.

Table 2 displays the breakdown of scoring across all SSP domains, in terms of typical, some difference, and definite difference. Table 2 shows that there were no significant associations between the 2–3 sff and 0–1 sff groups and any domain of the SSP.

Table 3 shows the VABS-II communication and socialisation sub-domains functional age equivalents (in months) and standard scores. Please note, that both *t*-tests and Mann–Whitney U tests were used depending on each subdomains meeting parametric assumptions. There were no significant differences between both groups in any VABS-II communication subdomains, or the communication standard score. With VABS-II socialisation scores (Table 3), there were no significant differences between the 2–3 sff and 0–1 sff groups on any of the socialisation subdomains or the socialisation standard score.

Table 3 shows the WISC-IV index scores. Those with 2–3 sff had a mean FSIQ score of 70.53, those with 0–1 sff had a mean FSIQ score of 82.77. Please note that in an average population FSIQ falls within the range of 90–109 [13]. Whilst a significant difference was found between those with 2–3 sff and 0–1 sff in terms of their VCI and FSIQ, after the Holm–Bonferroni correction, this was no longer significant. Therefore, there was no significant difference between the groups in WISC-IV index scores.

## 4. Discussion

The findings from this service evaluation suggest that when presenting to a clinical setting, those with FASD with few or no sentinel facial features present just as severely, in terms of central nervous system impairments, as those with more sentinel facial features. Whilst it has been long understood that those with clear sentinel facial features can present with severe neurocognitive deficits, this study helps add to the evidence that in a clinical sample, those without sentinel facial features can be just as severely affected.

When considering a range of measures, including underlying cognitive ability, sensory profiling, adaptive behaviour, and wider neurodevelopmental outcomes, there was no difference between the two groups. Therefore, our results suggest that the level of central nervous system impairment is not necessarily related to the number of sentinel facial features alone. There are various clinical and research implications based upon this. The comorbidity study by Popova et al. (2016), which identified 428 comorbid conditions were found to be diagnosed and associated as outcomes in people diagnosed with FASD [21], highlighted that FASD can present in different ways. As such, one hypothesis may be that, historically, the cases with clear sentinel facial features and more apparent central nervous systems deficits have been more easily identified in non-specialised clinical settings, which may well explain why previous studies have suggested that ‘severity’ is linked to facial features [10,22,23]. Whilst our findings do not in any way suggest that facial features are not important or that those with sentinel facial features are not “severe” in their presentation, they do suggest that those without sentinel facial features may be just as “severe” in a clinical sample. This may be due to differences in the National FASD Specialist Clinic cohorts compared to local samples or referral clinics in other countries. Nevertheless, when considering the wide prevalence of adaptive difficulties seen in the broader cohort, the finding of this service evaluation has significant implications on the levels of support required by those with any form of FASD, regardless of the number of facial features.

Numerous studies have identified that when looking for just Fetal Alcohol Syndrome, based primarily on facial features, cases of FASD are not often picked up, and prevalence rates are low [2]. Whilst FAS is not diagnosed just based on the sentinel facial features, in the UK at least, many clinicians rely on the facial features to decide on whether a person will even meet a threshold for clinical evaluation [24]. Therefore, the wider CNS profile is often missed and, in some cases, people are not even seen in a clinical setting. This has meant that the needs of these individuals are not supported and in the longer term, significant involvement with criminal justice, mental health services, and other missed support needs can occur [25]. The implication that there is no real difference, from our service evaluation at least, between the two groups highlights the need to identify individuals at risk early to prevent associated difficulties by using means other than assessing facial features.

In England, The National Institute for Health and Care Excellence (NICE) quality standards [26] have suggested that recording of alcohol exposure in all children is necessary. With prenatal alcohol exposure being the necessary risk factor leading to an FASD diagnosis, the fact that there is no significant difference between groups, yet those without the facial features are often unrecognised and missed, exacerbates their disabilities and the impact on both the individual but also on wider society. It is only by earlier recognition of CNS deficits that these trajectories can be changed. Therefore, it is essential that individuals with histories of exposure to alcohol and possible FASD should be offered a neuropsychological assessment in keeping with the recommendation in the new UK NICE quality standards [26].

Recognising that there is an increased vulnerability in this group is also an important factor for those who lack the obvious facial features. With increased exploitation of individuals with executive function and neurodevelopmental presentations, where there is no support, the impact of this vulnerability is exacerbated [25,27]. Further, the field of FASD is trying to move away from a deficit-based model of care to a strength-based model of support [28]. Early recognition is vital to this.

There has been increasing developments in support for both the individual and the family, which can now be directed to change trajectories. Consensus work around medication alongside newly developed therapeutic interventions, both psychological and environmental, are all being assessed for efficacy, meaning that understanding the individual and the difficulties is important. When there is still a focus on identifying FASD using only the facial features, many would miss out on support. Our work here would suggest that a more important area to focus on is evidence of alcohol exposure in pregnancy, alongside the adaptive and neuropsychological difficulties encountered. Whilst sentinel facial features and physical findings such as growth deficiency may continue to help identify those with prenatal alcohol exposure, especially where pregnancy information is missing, it would be wrong to suggest that those with all three sentinel facial features are on the more ‘severe’ end of the spectrum in relation to CNS deficits. Rather, they are perhaps more easily recognisable.

This work is not without its limitations. The present service evaluation had a sample with a broad age range (6–37 years) and the comparison groups significantly differed in terms of sex. Both are likely consequences of using a referral-based clinical cohort where cases were not screened prior to inclusion. The amount of data did not allow wider exploration to identify if this was a true finding or an incidental one. The data available did not allow full exploration of the subtleties of how timings of exposure would impact the cognitive and facial developments. Further, as a clinical sample, the primary aim is to see and assess the individuals. This means that, unlike a cohort study with set measures, the clinic will modify tools it uses and update approaches with changing evidence over time. Therefore, the number of people being exposed to tests in each group will vary, sometimes reducing the numbers available for comparison and introducing possible error. Where possible, this was minimised; however, it cannot be fully accounted for on all occasions based on the nature of the sample.

Whilst the findings also showed that alcohol exposure levels reported to our service seemingly were not leading to differences in terms of facial features, it is worth noting, however, that nearly all people attending the National FASD Specialist Clinic have moderate-to-high levels of prenatal alcohol exposure. Comment cannot be made about lower levels of alcohol exposure or different consumption patterns.

Future research should try to replicate this service evaluation but with greater control over the completion of assessments included. Additionally, as a UK national specialist clinic, those who are assessed are likely to have more significant need, requiring specialist support. Specifically, when compared to the population distribution of all those who would qualify for a diagnosis of ‘FASD without sentinel facial features’, our 0–1 sff group are likely to be more severely impacted in terms of CNS deficit, and thus our service evaluation is biased towards a null finding. There will be many people with FASD (either undiagnosed or diagnosed by other professionals) who were not available for inclusion in this service evaluation. Future work should strive to include those with FASD with varying level of need, adaptive skills, and cognitive abilities.

## 5. Conclusions

Whilst it remains important to consider sentinel facial features in the recognition of FASD, our data would suggest that there is an equal neuropsychological deficit between both groups, at least in a clinical population. Therefore, it is vital to assess and recognise the needs of all people with FASD and advocate support to be put in place to prevent subsequent harm, especially in those presenting with clinical need. This is to prevent associated difficulties, irrespective of the presence or absence of sentinel facial features.

## Figures and Tables

**Table 1 children-10-00266-t001:** Assessments and scores included in this service evaluation.

Assessments Included.	Assessment Scores Included
4-Digit Diagnostic Code [5]	Documented Level of Risk of Prenatal Alcohol Exposure; Unknown Risk, Some Risk, High Risk
Wechsler Intelligence Scale for Children, 4th Edition (WISC-IV) [13]	Verbal Comprehension Index (VCI), Perceptual Reasoning Index (PRI). Working Memory Index (WMI), Processing Speed Index (PSI), and Full-Scale IQ (FSIQ)
ADHD Screening Questionnaire (developed by the clinic, based on the DSM-IV)	Type of ADHD Diagnosis: Combined, Inattentive, Hyperactive-Impulsive, No Diagnosis Received
Social Communication Questionnaire (SCQ) [14]	Total SCQ Scores
Short Sensory Profile (SSP) [15]	All SSP domains: tactile sensitivity, taste/smell sensitivity, movement sensitivity, under responsive/ sensory seeking, auditory filtering, low energy/weak, visual/auditory sensitivity, total score. Scores are recorded as: Typical Performance, Probable Difference, Definite Difference.
Vineland Adaptive Behavior Scale, 2nd Edition (VABS-II) [16]	All Developmental Ages and Standard Scores from Communication and Socialisation Domains

**Table 2 children-10-00266-t002:** Comparisons of sex, comorbid diagnoses, assessment scores of risk of prenatal alcohol exposure, and sensory needs, between those with two or more sentinel facial features (2–3 sff) and those with one or fewer sentinel facial features (0–1 sff).

Diagnosis/Assessment	Subcategory/Subtest	Score Given	Groups		Statistics	
			2–3 sff *n* (%)	0–1 sff *n* (%)	*chi²*	*p*
Sex [2–3 *n* = 41, 0–1 *n* = 109]	Male		28 (68.3)	50 (45.9)	6.001	0.014
	Female		13 (31.7)	59 (54.1)		
ADHD Diagnosis [2–3 *n* = 41, 0–1 *n* = 108]	Combined		16 (39.0)	36 (33.3)	3.025	0.388
	Inattentive		16 (39.0)	50 (46.3)		
	Hyperactive-Impulsive		2 (4.9)	1 (0.9)		
	No Diagnosis Received		7 (17.1)	21 (19.4)		
ASD Diagnosis [2–3 *n* = 40, 0–1 *n* = 106]	ASD		31 (77.5)	77 (72.6)	0.356	0.551
	No Diagnosis Given		9 (22.5)	29 (27.4)		
4-Digit Score [2–3 *n* = 41, 0–1 *n* = 109]	Documented Level of Risk of Prenatal Alcohol Exposure	Unknown Risk	1 (2.4)	0 (0.0)	3.007	0.222
		Some Risk	19 (46.3)	46 (42.2)		
		High Risk	21 (51.2)	63 (57.8)		
SSP [2–3 *n* = 31, 0–1 *n* = 82]	Tactile Sensitivity	Typical	7 (22.6)	19 (23.2)	0.332	0.847
		Probable Difference	4 (12.9)	14 (17.1)		
		Definite Difference	20 (64.5)	49 (59.8)		
	Taste/Smell Sensitivity	Typical	19 (61.3)	47 (57.3)	1.667	0.435
		Probable Difference	3 (9.7)	16 (19.5)		
		Definite Difference	9 (29.0)	19 (23.2)		
	Movement Sensitivity	Typical	20 (64.5)	56 (68.3)	4.230	0.121
		Probable Difference	1 (3.2)	11 (13.4)		
		Definite Difference	10 (32.3)	15 (18.3)		
	Under Responsive/Sensory Seeking	Typical	3 (9.7)	12 (14.6)	0.515	0.773
		Probable Difference	1 (3.2)	2 (2.4)		
		Definite Difference	27 (87.1)	68 (82.9)		
	Auditory Filtering	Typical	2 (6.5)	4 (4.9)	2.456	0.293
		Probable Difference	0 (0.0)	6 (7.3)		
		Definite Difference	29 (93.5)	72 (87.8)		
	Low Energy/Weak	Typical	14 (45.2)	34 (41.5)	1.675	0.433
		Probable Difference	1 (3.2)	9 (11.0)		
		Definite Difference	16 (51.6)	39 (47.6)		
	Visual/Auditory Sensitivity	Typical	9 (29.0)	27 (32.9)	0.179	0.915
		Probable Difference	10 (32.3)	26 (31.7)		
		Definite Difference	12 (38.7)	29 (35.4)		
	Total Score	Typical	2 (6.5)	8 (9.8)	0.891	0.640
		Probable Difference	3 (9.7)	12 (14.6)		
		Definite Difference	26 (83.9)	62 (75.6)		

Abbreviations: 2–3 sff, 2–3 sentinel facial features; 0–1 sff, 0–1 sentinel facial features; ADHD, attention deficit hyperactivity disorder; ASD, autism spectrum disorder (or condition), SSP, Short Sensory Profile.

**Table 3 children-10-00266-t003:** Comparisons of assessment scores of social communication, communication and social adaptive skills, and cognition, between those with two or more sentinel facial features (2–3 sff) and those with one or fewer sentinel facial features (0–1 sff).

Assessment	Subtest	Groups		Statistics		
		2–3 sff Mean (SD) **Median (IQR)**	0–1 sff Mean (SD) **Median (IQR)**	*t*-test or Mann-Whitney U output	*p*	Corrected *p* value
Age, in years [2–3, *n* = 41, 0–1, *n* = 109]		**9.00 (7.00)**	**12.00 (6.00)**	*U* = 1629.5, *Z* = −2.560	0.010 *	0.070
SCQ [2–3, *n* = 34, 0–1 *n* = 89]		16.44 (7.90)	16.07 (6.07)	*t* (121) = −0.271	0.787	0.787
VABS-II	Communication—Receptive, age in months [2–3 *n* = 40, 0–1 *n* = 104]	**34.50 (25.50)**	**35.00 (26.25)**	*U* = 1905.0, *Z* = −0.781	0.435	1.000
	Communication—Expressive, age in months [2–3 *n* = 40, 0–1 *n* = 103]	**58.50 (23.00)**	**64.00 (30.00)**	*U* = 1717.5, *Z* = −1.542	0.123	0.738
	Communication—Written, age in months [2–3 *n* = 40, 0–1 *n* = 104]	**85.50 (26.25)**	**95.00 (31.75)**	*U* = 1801.0, *Z* = −1.243	0.214	0.856
	Communication Standard Score [2–3 *n* = 37, 0–1 *n* = 97]	**67.00 (13.50)**	**69.00** ** *(* ** **10.00** ** *)* **	*U* = 1679.0, *Z* = −0.576	0.565	1.000
	Socialisation—Interpersonal Relationships, age in months [2–3 *n* = 40, 0–1 *n* = 104]	43.10 (22.21)	52.98 (34.92)	*t* (142) = 1.831	0.069	0.414
	Socialisation—Play and Leisure, age in months [2–3 *n* = 40, 0–1 *n* = 104]	**46.50 (25.50)**	**44.50 (33.75)**	*U* = 1890.5, *Z* = −0.846	0.398	1.000
	Socialisation—Coping Skills, age in months [2–3 *n* = 40, 0–1 *n* = 103]	**46.50 (38.75)**	**55.00 (37.00)**	*U* = 1736.0, *Z* = −1.458	0.145	0.738
	Socialisation Standard Score [2–3 *n* = 36, 0–1 *n* = 97]	61.33 (14.83)	64.59 (11.91)	*t* (131) = 1.311	0.192	0.608
WISC-IV	VCI [2–3 *n* = 20, 0–1 *n* = 58]	76.30 (16.85)	85.48 (14.63)	*t* (76) = 2.327	0.023 *	0.184
	PRI [2–3 *n* = 21, 0–1 *n* = 51]	82.04 (15.14)	88.45 (17.73)	*t* (70) = 1.450	0.152	0.608
	WMI [2–3 *n* = 21, 0–1 *n* = 54]	75.61 (17.54)	82.27 (15.12)	*t* (73) = 1.636	0.106	0.530
	PSI [2–3 *n* = 19, 0–1 *n* = 56]	81.42 (18.40)	86.91 (16.18)	*t* (73) = 1.234	0.221	0.608
	FSIQ [2–3 *n* = 15, 0–1 *n* = 36]	70.53 (17.45)	82.77 (17.85)	*t* (49) = 2.245	0.029 *	0.203

* *p* < 0.05; Abbreviations: 2–3 sff, 2–3 sentinel facial features; 0–1 sff, 0–1 sentinel facial features; SCQ, Social Communication Questionnaire; VABS-II, Vineland Adaptive Behavior Scale–2nd Edition; WISC-IV, Wechsler Intelligence Scale for Children—4th Edition; VCI, Verbal Comprehension Index; PRI, Perceptual Reasoning Index; WMI, Working Memory Index; PSI, Processing Speed Index; FSIQ, Full Scale IQ.

## Data Availability

Clinical data is held by the FASD clinic in keeping with NHS data storage rules. It is not openly available due to consent to share issues of the NHS however please contact lead author for more information.

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
