# Peer review of "Evaluating the Difference in Neuropsychological Profiles of Individuals with FASD Based on the Number of Sentinel Facial Features: A Service Evaluation of the FASD UK National Clinic Database"

_children, 2023, doi:10.3390/children10020266_

Round 1

Reviewer 1 Report

The authors breakdown a cohort of people with FASDs by number of cardinal facial features and compare outcomes of various behavior tests between people with 0 or 1 feature and those with 2 or 3 features. Contrary to what many may expect, there are no differences between the groups thus suggesting that facial features do not necessarily predict brain-related effects of prenatal alcohol exposure. The study has limitations, many are acknowledged in the Discussion, and the paper adds to the body of knowledge around FASD.

Comments for improvements

Include information about age range of participants for each test.

Include information about what a “normal” score would be on these tests so that the reader can put these findings in context.

Consider that the timing of eye development and lip development do not completely overlap and so the people with 1 or 2 features may provide insight into timing of alcohol exposure. Would the authors expect that people with only lip features might be different to people with lip and eye features?  

Abstract should include a note about the high comorbidity with ASD and ADHD rather than just noting this “was reviewed”.

Table 2 is hard to read – particularly for the 4-Digit Score and SSP.

In Table 3 move the Z onto the same line as its score.

Author Response

Reviewer 1

Include information about age range of participants for each test.

Thank you for the comment. Unfortunately, we are unable to provide this information as it is not on our database.

Include information about what a “normal” score would be on these tests so that the reader can put these findings in context.

Thank you for the comment. We’ve added comment to methods that describe normal values and where they are obtained

Consider that the timing of eye development and lip development do not completely overlap and so the people with 1 or 2 features may provide insight into timing of alcohol exposure. Would the authors expect that people with only lip features might be different to people with lip and eye features?  

Thank you for the comment. This is not something we were able to explore but have added it to the discussion and the limitations of the study.

Abstract should include a note about the high comorbidity with ASD and ADHD rather than just noting this “was reviewed”.

Thank you for the comment. This has been addressed.

Table 2 is hard to read – particularly for the 4-Digit Score and SSP.

Thank you for the comment. This has been addressed, by adding spaces to separate between assessment subtests and adding lines to the table to separate assessments.

In Table 3 move the Z onto the same line as its score.

Thank you for the comment. This has been changed.

Reviewer 2 Report

This is an excellent paper which has true significance for the FASD field. It adds to the discussions related to inappropriate presumptions based on facial sentinel features. It also highlights the importance of effective screening. The paper takes care to show the strengths and limits of the research. 

Author Response

Reviewer 2

No specific comments made

Reviewer 3 Report

This paper describes an interesting study conducted as an exploratory service evaluation audit performed in the UK National FASD Clinic to explore whether there are differences in the CNS profile based on the number of sentinel facial features present. The authors state there is an “assumption” in the previous literature that those with all three sentinel facial features present more “severely”, and this “assumption” is not consistent with “anecdotal impressions” obtained from clinicians from the UK National FASD Clinic. This study compares whether CNS deficits in cognition, adaptive functioning, and social communication are similar or different in clinic-referred alcohol-exposed individuals with varying numbers of the sentinel facial features. The sample is composed of 150 individuals referred to the National FASD clinic who have suspected fetal alcohol exposure. This clinic completes a comprehensive assessment including a dysmorphology assessment and extensive behavioral and cognitive testing to determine FASD diagnosis. Due to sample size, comparisons on the WISC/WAIS, adaptive behavior, and social communications were performed on alcohol-exposed individuals with either 2 or more sentinel facial features (PFL < 3rd %ile, smooth philtrum, thin upper lip) (= 41) vs. those with 1 or no facial features (= 109) with ages varying between 6 and 37 years. The authors chose to cut the number of sentinel features at 2 rather than at 3, given that some of the literature and their analyses showed no difference between those with 2 vs. 3 sentinel facial features. Results indicated that there was a biological sex imbalance between the two groups and that the group with 0-1 facial features were significantly older. (Throughout the paper, the word “gender” should be changed to “sex,” which the APA says should be used to refer to biological sex, whereas “gender” refers to environmental/cultural differences.) Comorbid diagnoses of autism spectrum disorders and ADHD were similar between the two groups, and no differences in sensory processing, communication, and adaptive behaviors were found. Those with 2-3 facial features had significantly lower Full Scale IQ and Verbal Comprehension IQ, but these were no longer significant after applying a p-value correction for multiple comparisons. The authors conclude that in this clinical population, there is “equal neuropsychological deficit” between dysmorphia groups and thus advocate that it is important to recognize the need for specialist intervention for those lacking the obvious facial features of prenatal alcohol exposure.

This study reports a null finding, but a number of major changes to the manuscript are warranted to clarify the aims of the study and the recommendations the authors want to make based on these findings. These details are necessary to show why the publication of these null findings are justified and would add to the literature. Additionally, extensive editing is required throughout the paper to make the language more direct and less convoluted.

Title—

The title of the paper does not accurately characterize the analyses reported in the paper: “Evaluating the difference in neuropsychological profiles of individuals with FASD based on the number of sentinel facial features…” The title refers to an analysis using continuous number of sentinel features rather than the dichotomous comparisons that were presented.

This error was also made in the first sentence of the Discussion: “The findings from this study suggest that the number of sentinel facial features does not appear to be associated with the presence of the central nervous system impairments 220 in those with a FASD.”

Introduction--

The paper begins with the statement that “Fetal Alcohol Spectrum Disorder (FASD) is a diagnostic term…” Actually, FASD is not considered a diagnosis in the US. The diagnoses accepted everywhere are FAS, pFAS, and ARND.

What is a “service evaluation audit” and how might this audit affect the collection, blindness, etc., of the study?

The stated goal of the paper, to examine “whether there are differences in the profile of CNS deficits based on the varying numbers of sentinel facial features,” is an excellent question and of much interest. The assessments included in the study (listed in Tabe 1) are appropriate and comprehensive. As noted above, the major problem, however, is the cut-off chosen by the authors.

The Introduction should be expanded to make clearer why this comparison between those with 2 or more facial features vs. those with 0-1 features is necessary and, as noted before, why they used a cut-off of 2 rather than 3 to test their hypothesis. Actually, the analyses need to be changed to compare those with 3 features vs. 2 or fewer. In addition, is there evidence from this clinic or other studies that those with more sentinel facial features receive more or better specialized care and interventions than those with less or none of the facial features?

Line 35--“UK National Clinic” Please use the official full name of the clinic here and throughout the rest of the paper, be consistent in how you refer to the clinic. For example, in other parts of the paper it is referred to as “UK national clinic”, “UK National FASD Clinic”, “The National FASD clinic”, “UK Clinic”, “National specialist clinic”.

Lines 54-56--I disagree that “little research has been done comparing CNS abilities in those with and without sentinel facial features of FASD”. Most studies involving the range of FASD outcomes usually do compare groups with and without dysmorphia on cognitive function, brain structure, and behavior, albeit the findings between studies are often inconsistent. The authors should discuss some of these findings and note the inconsistency.

Lines 57-59--Is it really just an assumption that those with all three features present more “severely”? I would say there is evidence that those with dysmorphia tend to present CNS deficits more severely (for example, see Ervalahti et al., 2007; Roussotte et al., 2012; Suttie et al, 2013; 2018).

Line 69--Define the acronyms ASD and ADHD, when first used. This was first mention of them in the paper.

Methods—

The range 6 to 37 years is too broad to test the questions raised in the paper, given major changes in facial features occurring during this age range. Perhaps the authors could break the groups into different age groups (child, adolescent, young adult, adult).

Lines 75-77--Over which years were the participants included in the analyses assessed? Be more specific than the “last decade”. It would be good to also state here how many participants will be analyzed. Other than those who declined to use their data for evaluations, were any other participants excluded from the analyses for other reasons?

Line 79--The word “specialist” should be changed to “specialized”.

Lines 80-81-- “Referrals are received from all parts of the UK funded by the NHS or on an individual case-by-case basis.” This sentence is confusing and should be clarified/more specific.

Lines 83-84--How long was the “gap” between testing days? Provide a range, if possible.

Lines 88-90--This following sentence can be better worded: “The clinic is one of the few specialist places in the UK that CAN reliably and accurately DIAGNOSE FASD using set diagnostic standards, without sentinel facial features being present.”

Lines 94-96--Were the examiners who administered the cognitive functioning tasks trained to identify the sentinel facial features? If so, could this have led to examiner bias in test administration? This is important.

Lines 128-129--The Revised IOM guidelines (2016) allow for the presence of at least 2 of 3 facial features for the diagnosis of FAS or pFAS. Please correct.

Lines 143-146--Can more details be provided as to how the sample was composed of 150? Were the 150 all the cases seen at the clinic who consented during the study years? Were others who consented excluded from the analyses for any reason?  Also mention which years this study covers.

Results--

Add mean/median ages and comparison test to Table 2. Same for sex, add n’s and percentages to Table 2. M’s, N’s, X2, t, and p should all be italicized; some of these are italicized but not consistently.

Lines 162-163--Describe what the sex imbalance was. Provide percentages for the sexes for each group. It would be good to add this sex data to Table 2.

Lines 163-166--The group with 0-1 features was significantly older than the group with 2-3 features, and there was a wide range of ages in both groups. Was there a correlation between age and the prevalence of each of the features? Previous work has found that facial dysmorphology assessments may be less sensitive at older ages and there is less prevalence of facial features in older ages (see Jacobson et al., 2021).

Line 175--Instead of repeating the p-value for the ADHD 4-group chi-square that is displayed in the table, perhaps report a chi-square and p-value for ADHD (yes/no) vs. the two facial feature groups.

Line 177-178--“…which is indicative of ASD.” Please state the precise terminology that the SCQ uses for a cut-off of 15.

Lines 188-191--The p-values do not need to be repeated, they are displayed in Table 2.

Lines 194-201--How do any of these have a p-value of 1.000? Please describe what about the distributions of the VABS sub-domains warranted some of them to be analyzed with a t-test and other with a Mann-Whitney U-test?

Table 2--SSP analyses. It appears that most of the “Probable Difference” categories have low N’s. Can the categories “Probable Difference” and “Definite Difference” be combined, and those data presented instead? It would make the Table a lot easier to read.

Lines 202-208--No significant differences were detected likely due to sample size issues and an overbearing p-value correction. However, there definitely does appear to be a functionally significant difference in terms of FSIQ. FSIQ was 12 points (almost 1 SD lower) in the group with 2-3 features. Also, the t-values and p-values that are displayed in Table 3 do not need to be repeated here.

Lines 209-213--Not clear why there is a separate analysis of WAIS given how few were analyzed (= 16). This analysis does not add to the paper and just increases the number of comparisons that are being corrected for. Either remove the WAIS analyses or combine with WISC scores since the scale of the IQ scores are the same between the two tests.

Discussion--

This study recommends that “…it is vital to assess, recognise the needs of all people with FASD and advocate support to be put in place to prevent subsequent harm. This is to prevent associated difficulties, irrespective of the presence or absence of sentinel facial features.” It is not clear that this is a problem in the clinical setting. Can more evidence be provided showing that clinic-referred exposed individuals without facial features are less likely to receive specialized support? Is this a problem at the UK National clinic?

Lines 223-231--These passages are hard to understand and should be re-worded more clearly. It is also important to mention the numerous studies that have found and not merely suggested that there can be differences in “severity” between diagnostic groups.

Line 234: “did not suggest   this link…”—too much space between words.

Lines 240-241: “Numerous studies have identified that when looking for just fetal alcohol syndrome, based primarily on facial features, cases of FASD are not often picked up, and prevalence rates are low [2].”  A diagnosis of FAS requires not only identifying the three sentinel features but also diminished growth and behavior. For pFAS, some information about a history of maternal drinking is necessary. Otherwise, the clinician may ascribe FAS to a disorder that mimics some of the FAS features (Hoyme et al., 2005).

Author Response

Reviewer 3

Title—

The title of the paper does not accurately characterize the analyses reported in the paper: “Evaluating the difference in neuropsychological profiles of individuals with FASD based on the number of sentinel facial features…” The title refers to an analysis using continuous number of sentinel features rather than the dichotomous comparisons that were presented.

This error was also made in the first sentence of the Discussion: “The findings from this study suggest that the number of sentinel facial features does not appear to be associated with the presence of the central nervous system impairments 220 in those with a FASD.”

Thank you for the comment. We accept that we eventually had dichotomous groups, but our original groups were based on continuous data, later being categorised for analysis purposes. We do still believe the title reflects this approach.

Thank you again for the comment, please see above.

Introduction--

The paper begins with the statement that “Fetal Alcohol Spectrum Disorder (FASD) is a diagnostic term…” Actually, FASD is not considered a diagnosis in the US. The diagnoses accepted everywhere are FAS, pFAS, and ARND.

What is a “service evaluation audit” and how might this audit affect the collection, blindness, etc., of the study?

The stated goal of the paper, to examine “whether there are differences in the profile of CNS deficits based on the varying numbers of sentinel facial features,” is an excellent question and of much interest. The assessments included in the study (listed in Tabe 1) are appropriate and comprehensive. As noted above, the major problem, however, is the cut-off chosen by the authors.

The Introduction should be expanded to make clearer why this comparison between those with 2 or more facial features vs. those with 0-1 features is necessary and, as noted before, why they used a cut-off of 2 rather than 3 to test their hypothesis. Actually, the analyses need to be changed to compare those with 3 features vs. 2 or fewer. In addition, is there evidence from this clinic or other studies that those with more sentinel facial features receive more or better specialized care and interventions than those with less or none of the facial features?

Line 35--“UK National Clinic” Please use the official full name of the clinic here and throughout the rest of the paper, be consistent in how you refer to the clinic. For example, in other parts of the paper it is referred to as “UK national clinic”, “UK National FASD Clinic”, “The National FASD clinic”, “UK Clinic”, “National specialist clinic”.

Lines 54-56--I disagree that “little research has been done comparing CNS abilities in those with and without sentinel facial features of FASD”. Most studies involving the range of FASD outcomes usually do compare groups with and without dysmorphia on cognitive function, brain structure, and behavior, albeit the findings between studies are often inconsistent. The authors should discuss some of these findings and note the inconsistency.

Lines 57-59--Is it really just an assumption that those with all three features present more “severely”? I would say there is evidence that those with dysmorphia tend to present CNS deficits more severely (for example, see Ervalahti et al., 2007; Roussotte et al., 2012; Suttie et al, 2013; 2018).

Line 69--Define the acronyms ASD and ADHD, when first used. This was first mention of them in the paper.

Thank you for the comment, however in the UK FASD has been a diagnostic term since 2019. We have made a change in the paper to reflect this.

Thank you for the comment. To clarify, this work is a service evaluation, not a research study. This is a clinical sample, offering a naturalistic evaluation of a presenting population, rather than a blinded population sample. The word ‘audit’ was used in error, this has been rectified, and is consistent throughout.

Thank you for the comment but I would disagree with the reviewer here. There is international dispute as to what the cut off should be as the reviewer is clearly aware of. The cut offs and an exploration of this is included in the methods and the discussion. There may be limitations but we do believe the cut offs have been adequately explored and will hopefully lead others to loot at similar questions using the cut offs from other approaches.

Within scope of limitations of word count and conciseness, we have briefly mentioned rationale behind this in introduction and expanded on this in the method section, which we believe address comment made. Expanding in the introduction would duplicate the rationale within the methods section and approach taken. We did do analysis (as stated in methods) of 3 vs 2 or fewer, and because of numbers of cases we decided to combine them, this was in part a pragmatic decision to increase power of the statistical analysis. This is mentioned in the methods section.

Thank you for the comment. This has been changed.

Thank you for the comment. I have amended this. I think we are not trying to imply those with sentinel features are not severe in the presentation rather the opposite that those without features may be just as severe, especially when presenting to a clinical sample which has implication for practice. I think that may not have been as clear and we have tried to make this more obvious

Thank you for this comments. As per the comment above we have tried to make this clearer. Further at this point in the paper we are beginning to explore that and do state that the finding that the presentation is more severe could be a correct one.

Thank you for the comment. This has been changed.

Methods—

The range 6 to 37 years is too broad to test the questions raised in the paper, given major changes in facial features occurring during this age range. Perhaps the authors could break the groups into different age groups (child, adolescent, young adult, adult).

Lines 75-77--Over which years were the participants included in the analyses assessed? Be more specific than the “last decade”. It would be good to also state here how many participants will be analyzed. Other than those who declined to use their data for evaluations, were any other participants excluded from the analyses for other reasons?

Line 79--The word “specialist” should be changed to “specialized”.

Lines 80-81-- “Referrals are received from all parts of the UK funded by the NHS or on an individual case-by-case basis.” This sentence is confusing and should be clarified/more specific.

Lines 83-84--How long was the “gap” between testing days? Provide a range, if possible.

Lines 88-90--This following sentence can be better worded: “The clinic is one of the few specialist places in the UK that CAN reliably and accurately DIAGNOSE FASD using set diagnostic standards, without sentinel facial features being present.”

Lines 94-96--Were the examiners who administered the cognitive functioning tasks trained to identify the sentinel facial features? If so, could this have led to examiner bias in test administration? This is important.

Lines 128-129--The Revised IOM guidelines (2016) allow for the presence of at least 2 of 3 facial features for the diagnosis of FAS or pFAS. Please correct.

Lines 143-146--Can more details be provided as to how the sample was composed of 150? Were the 150 all the cases seen at the clinic who consented during the study years? Were others who consented excluded from the analyses for any reason?  Also mention which years this study covers.

Thank you for the comment, we acknowledge this is potentially a limitation that reflects the nature of the sample taken from the clinic. We have acknowledged this in the limitations.

Thank you for the comment. This has been addressed.

Thank you for the comment. This has been changed.

Thank you for the comment. This has been addressed and clarified that ‘referrals are received from all parts of the UK, and assessments are funded by NHS…’.

Thank you for the comment. This has been added – approximately 3 months.

Thank you for the comment. This has been changed.

Thank you for the comment. No, the clinicians who assess cognitive functioning are experienced in working with this population but are not trained in identifying facial features. Identifying facial features/using the assessment software is carried out by the FASD Specialist Clinician. This has been addressed in the report.

Thank you for the comment. We have reworded this, one of the rationale for considering 2 features and 3 features was because of the IOM allowance for this in our initial analysis.

Thank you for the comment. Of those seen in clinic, the first 150 were included in this analysis. The database is ongoing and continues to develop. Only those who have not consented, are excluded from being added to the database. Those who were assessed but did not receive a confirmed diagnosis of FASD were not included in this analysis.

Results--

Add mean/median ages and comparison test to Table 2. Same for sex, add n’s and percentages to Table 2. M’s, N’s, X2, t, and p should all be italicized; some of these are italicized but not consistently.

Lines 162-163--Describe what the sex imbalance was. Provide percentages for the sexes for each group. It would be good to add this sex data to Table 2.

Lines 163-166--The group with 0-1 features was significantly older than the group with 2-3 features, and there was a wide range of ages in both groups. Was there a correlation between age and the prevalence of each of the features? Previous work has found that facial dysmorphology assessments may be less sensitive at older ages and there is less prevalence of facial features in older ages (see Jacobson et al., 2021).

Line 175--Instead of repeating the p-value for the ADHD 4-group chi-square that is displayed in the table, perhaps report a chi-square and p-value for ADHD (yes/no) vs. the two facial feature groups.

Line 177-178--“…which is indicative of ASD.” Please state the precise terminology that the SCQ uses for a cut-off of 15.

Lines 188-191--The p-values do not need to be repeated, they are displayed in Table 2.

Lines 194-201--How do any of these have a p-value of 1.000? Please describe what about the distributions of the VABS sub-domains warranted some of them to be analyzed with a t-test and other with a Mann-Whitney U-test?

Table 2--SSP analyses. It appears that most of the “Probable Difference” categories have low N’s. Can the categories “Probable Difference” and “Definite Difference” be combined, and those data presented instead? It would make the Table a lot easier to read.

Lines 202-208--No significant differences were detected likely due to sample size issues and an overbearing p-value correction. However, there definitely does appear to be a functionally significant difference in terms of FSIQ. FSIQ was 12 points (almost 1 SD lower) in the group with 2-3 features. Also, the t-values and p-values that are displayed in Table 3 do not need to be repeated here.

Lines 209-213--Not clear why there is a separate analysis of WAIS given how few were analyzed (= 16). This analysis does not add to the paper and just increases the number of comparisons that are being corrected for. Either remove the WAIS analyses or combine with WISC scores since the scale of the IQ scores are the same between the two tests.

Thank you for the comment, we have tried to make appropriate changes.

Thank you for the comment. We have added the sex information to the table which will help to address this comment. This is further commented on in the discussion.

Thank you for the comment. That information could not be extracted from this clinical database. The benefit of database like this, is to develop naturalistic hypotheses which can be further tested in the future.

Thank you for the comment. It is important to differentiate the different subtypes of ADHD as this is important to the behavioural management of each group.

Thank you for the comment. This has been addressed.

Thank you for the comment. This has been changed.

Thank you for the comment. The p-value of 1.000 are a result of the Holm-Bonferroni correction. Both t-tests and Mann-Whitney U-tests were used as a result of on the subdomains meeting parametric assumptions. A clarifying sentence has been added to the results section.

Thank you for the comment. We have decided not to combine ‘probably and definite difference’ as they demonstrate different unique sensitives of the groups.  

Thank you for the comment. People without facial features can present just as severely in a clinical sample – needs to be made more explicit. Whilst there will be some differences, these will be minimal, therefore it’s important to focus on all people with neuropsychological deficits rather than just facial features.

Thank you for the comment. This has been changed.

Thank you for the comment. The WAIS has been removed to reduce confusion. As a clinical sample we see people of all ages and have to use the appropriate test, the data presented reflects that.

Discussion--

This study recommends that “…it is vital to assess, recognise the needs of all people with FASD and advocate support to be put in place to prevent subsequent harm. This is to prevent associated difficulties, irrespective of the presence or absence of sentinel facial features.” It is not clear that this is a problem in the clinical setting. Can more evidence be provided showing that clinic-referred exposed individuals without facial features are less likely to receive specialized support? Is this a problem at the UK National clinic?

Lines 223-231--These passages are hard to understand and should be re-worded more clearly. It is also important to mention the numerous studies that have found and not merely suggested that there can be differences in “severity” between diagnostic groups.

Line 234: “did not suggest   this link…”—too much space between words.

Lines 240-241: “Numerous studies have identified that when looking for just fetal alcohol syndrome, based primarily on facial features, cases of FASD are not often picked up, and prevalence rates are low [2].”  A diagnosis of FAS requires not only identifying the three sentinel features but also diminished growth and behavior. For pFAS, some information about a history of maternal drinking is necessary. Otherwise, the clinician may ascribe FAS to a disorder that mimics some of the FAS features (Hoyme et al., 2005).

Thank you for the comment. We want to highlight that not everywhere has access to full neuropsychological testing and speaking anecdotally this testing is restricted to those with sentinel facial features. We want to demonstrate it is important this testing is done to everyone with this FASD presentation, regardless of facial features.

Thank you for the comment. As per the comment above we have tried to make it clearer here

Thank you for the comment. This has been changed.

Thank you for the comment. As per the comment above we have tried to make it clearer here and make our point clearer.

Throughout the paper, the word “gender” should be changed to “sex,” which the APA says should be used to refer to biological sex, whereas “gender” refers to environmental/cultural differences

Thank you for the comment. This has been changed.

Round 2

Reviewer 3 Report

I continue to think that the title of the paper should be revised, as originally noted. 

This is an interesting study but the reporting of this null finding was not convincing based on the analyses conducted. Few substantive changes seem to have been seriously considered.